# Strategies of Predictive Schemes and Clinical Diagnosis for Prognosis Using MIMIC-III: A Systematic Review

**DOI:** 10.3390/healthcare11050710

**Published:** 2023-02-27

**Authors:** Sarika R. Khope, Susan Elias

**Affiliations:** School of Electronics Engineering, Vellore Institute of Technology, Chennai 600127, Tamil Nadu, India

**Keywords:** predictive, machine learning, deep learning, MIMIC-III, ICD9, critical care

## Abstract

The prime purpose of the proposed study is to construct a novel predictive scheme for assisting in the prognosis of criticality using the MIMIC-III dataset. With the adoption of various analytics and advanced computing in the healthcare system, there is an increasing trend toward developing an effective prognostication mechanism. Predictive-based modeling is the best alternative to work in this direction. This paper discusses various scientific contributions using desk research methodology towards the Medical Information Mart for Intensive Care (MIMIC-III). This open-access dataset is meant to help predict patient trajectories for various purposes ranging from mortality forecasting to treatment planning. With a dominant machine learning approach in this perspective, there is a need to discover the effectiveness of existing predictive methods. The resultant outcome of this paper offers an inclusive discussion about various available predictive schemes and clinical diagnoses using MIMIC-III in order to contribute toward better information associated with its strengths and weaknesses. Therefore, the paper provides a clear visualization of existing schemes for clinical diagnosis using a systematic review approach.

## 1. Introduction

It is a general observation that the moment a patient is admitted to a hospital for any form of treatment, various detailed information is recorded for the entire stay and treatment of the patient. This information has been retained for over a decade and is used for further analysis, service investigation, diseases, etc. Over time, a healthcare unit accumulates more significant and massive amounts of such data, which acts as clinical evidence [1]. Healthcare professionals generally study all this to offer the most constructive treatment plans [2]. If such massive data were manually assessed, it would consume tedious effort and valuable time, but it would contribute to a better prognosis system [3]. If a patient’s profile information is sequentially structured, all this information will yield a better prognosis. This will further contribute to formulating an effective treatment plan and simultaneously chalk out different preventive mechanisms [4]. It can also identify essential parameters that can be used for selecting alternative medicine, adopting new treatment plans or changes in services, or taking a precise decision to reduce the mortality rates during the hospital stay post treatment. This information about the patient also helps the health insurance organization forecast the client’s health statistics in order to offer them better insurance plans. In this regard, the present-day hospitals generally use an Electronic Health Record (EHR) system where all patient information management is carried out [5]. An EHR system also generates a massive amount of clinical data stored in cloud-based services and performs various analytics [6,7]. From this perspective, ‘disease progression’ is one of the emergent topics among healthcare professionals, where the prediction of consequences associated with the current treatment plan is carried out [8]. This is also termed as prediction of the patient’s trajectory. The patient’s sequential information based on temporal factors is gathered, and learning algorithms are applied to forecast the following possibilities of an event.

The term “event” means the incident after which the patient is admitted to a hospital to undergo a specific treatment. The term also refers to a different course of treatment; the medication given, outcomes of all forms of diagnostics, etc. This information is so massive that it is pretty challenging to study. Therefore, they are encoded by a globally recognized standard called the International Statistical Classification of Diseases and Related Health Problems, i.e., ICD-9, where the number nine means the ninth revision. It has also been observed that machine learning is one of the most prominent approaches for working toward predictive analysis [9,10]. Due to this capacity of classifying and performing predictions, deep learning has triggered the interest of many. However, implementing deep learning does not always solve the problem, although there are various literature archives with reported success factors.

This paper presents an insight into the effectiveness of various existing predictive approaches over the MIMIC-III dataset to identify open research issues. The contributions of the study are as follows:The study discusses significant challenges associated with the MIMIC-III dataset.An elaborative and concise review of predictive methods applied to the MIMIC-III dataset are carried out regarding classification, a standard predictive approach, and an early predictive approach.The study addresses various techniques used for clinical diagnosis of the MIMIC-III dataset, including disease characterization interpretation, detection decision approach, and evaluation-based scheme.In addition, the study examines the significant learning outcomes in terms of the strengths and weaknesses of existing methods to analyze the MIMIC-III dataset.The study also highlights essential open-ended issues.

Therefore, the prime objective of the proposed study is to have a critical insight towards various computational analytical approaches using the MIMIC-III dataset with an agenda to understand the strengths and weaknesses of existing techniques towards facilitating clinical diagnosis.

## 2. Methodologies

The proposed review work is discussed using a PRISMA diagram as exhibited in Figure 1 where the rationale for identification, screening, eligibility, and information about the included journals have been presented. The primary selection criteria of journals are mainly based on its inclusion towards a computational model that has used the MIMIC-III dataset as there are various clinical-based studies being carried out using similar dataset. The agenda mainly focuses on evaluation of strengths and weaknesses of existing computational approaches.

Despite the increasing adoption of the MIMIC-III dataset, there needs to be a clarification about the strengths and weaknesses associated with various analytical levels of implementation connected to it. Therefore, the primary concern of the adopted research methodology is to study the research papers that have implemented their strategies of clinical analysis over the MIMIC-III dataset [11]. The overall flow of adopted methodology is showcased in Figure 2, which represents the start and inclusion of all essential processes being carried out towards reviewing the research articles with an agenda to understand the effectiveness of existing analytical approaches in the MIMIC-III dataset.

The idea is to understand various possible degrees of research-based works for diagnosing critical illness by adopting the MIMIC-III dataset. The detailed information of adopted PRISMA-based methodology in the present work is as follows:Type of Research: A desk research methodology is adopted that mainly infers information based on information available from reliable internet sources and archives of scientific concepts [12]. This method uses aggregated data followed by summarized data to leverage the cumulative investigational effectiveness. This process of desk research methodology can be further elaborated from the adopted bibliographic research viewpoint considering the domain of research work [13].Data Collection Method: The proposed scheme constructs a bibliography from varied sources of technical and experimental documents and articles [14]. The proposed study uses the Google search engine to find the names of all reputed Q1 and Q2 publishers and does not use the Google search engine to find the direct manuscript owing to the irrelevant number of hits. Using a Google search engine, the study identifies various relevant and top-notch journals to find the proposed topic. For, e.g., IEEE Transaction on Biomedical Engineering, IEEE Transactions on Industrial Informatics, IEEE/ACM Transactions on Computational Biology and Bioinformatics, BMC Medical Informatics and Decision Making (Springer), BMC Journal of Translational Medicine (Springer), International Journal of Medical Informatics (Elsevier), etc. The proposed study adopts PRISMA guidelines to construct its search strategy. The first author Sarika R. Khope has consulted journals such as IEEE, BMC, Elsevier, and MDPI, while the second author Susan Elias has referred to journals such as PubMed, arXiv, SAGE, and Springer. While performing the primary search using the above-mentioned query keywords during the period of 2013–2023, there are a total of 174,332 records. Finally, after applying a filter for the last 5 years, a total of 23,533 records have been obtained, which is finally subjected to filtering where EndNote has been used to remove the duplicates, and this has resulted in 1106 records as exhibited in Table 1.

Within this journal, the individual manuscript has been searched using the keywords “Machine learning techniques for MIMIC-III”, “prediction methods for MIMIC-III”, and “disease diagnosis using MIMIC-III”. In addition, various keywords, e.g., “critical disease diagnosis with MIMIC-III”, “healthcare data analysis in MIMIC-III”, “intensive care unit and MIMIC-III” were also tried to check if some vital implementation study had been skipped while aggregating study models over the MIMIC-III dataset. The proposed system uses ROBIS, a tool to carry out assessment of risk of bias towards systematic review. The steps adopted towards reducing bias are as follows: (i) prior to start of data collection, a set of research questions have been formulated in order to accomplish a clear objective towards the systematic review; (ii) multi-level evaluation of inclusion and exclusion criteria; (iii) addressing bias while performing study selection followed by its synthesis. The proposed system uses inclusion and exclusion criteria where a total of 265 preliminary implementation papers have been finally filtered into 106 manuscripts to draft this review work. The search sources for retrieving bibliographic database are: ICU, ICD-9, prediction, mortality, learning, disease, patient, health records, classification, critical care. Following are the inclusion and exclusion criteria:○Inclusion Criteria: All the filtered study papers should discuss or offer a potential guideline toward design/algorithmic implementation approach using the MIMIC-III dataset. The keywords for inclusion criteria are: (i) primary keyword: MIMIC-III, (ii) secondary key word: result analysis, numerical analysis, experimental data, mathematical modelling, algorithm discussion.○Exclusion Criteria: Any publication before 2017 has not been considered in the presented study. In addition, papers related to theoretical and iterative discussions are avoided. The keywords for exclusion criteria are: (i) primary keyword: MIMIC II, publication year less than 2017; (ii) secondary keyword: review, survey without significant insights.

Out of 23,533 records, 841 records have been excluded after reviewing title and abstract, which leads to a total of 1106 records using inclusion and exclusion criteria discussed above. Further, 841 records have been excluded resulting in the arrival of 265 records. In the next round of filtering, further 150 articles or records have been excluded after reading the whole paper to finally arrive at 106 articles. The prime reason of exclusion was lack of solid computational model/architecture, non-algorithmic approaches, and lack of mathematical base to understand and justify the outcome.

Constructing Structure of Manuscript Discussion: The MIMIC-III dataset is mainly used to analyze critical information during the hospital stay. Hence, various applications in this regard are connected to multiple predictive approaches [15]. It is also noted that machine learning and deep learning are the core technologies used in this perspective [16,17]. Therefore, the paper discusses classification and the usual predictive approach followed by an early predictive approach that covers most of the existing trends in bibliographic research. On the other hand, there are also various approaches to clinical diagnosis in order to offer more clinical insight. Hence, the paper discusses disease characterization-based techniques, detection techniques, and evaluation-based techniques associated with disease diagnosis for the MIMIC-III Dataset. This assists in offering a better flow of discussion toward assessing the impact of all existing research-based implementation models on disease diagnosis on the MIMIC-III dataset.

## 3. Significance/Challenges of MIMIC-III

MIMIC-III is a publicly available dataset that consists of large amounts of health-related information from 2001 to 2012; Beth Israel Deaconess Medical Center collected this dataset information [18]. It includes vast information associated with the measurement of vital signs extracted during the stay of the patient in the medical center, demographics of the respective patient, different information about the procedure being performed, the result of laboratory tests, prescribed medicines, clinical suggestions, radiology reports, as well as specific details on mortality. Currently, the MIMIC-III dataset is highly in demand for performing analysis of acute diseases ranging from enhancing the decision rule, epidemiology, developing the electronic tool, etc. It also possesses information about temporal data during the entire duration of the stay in the hospital. The data is stored in a tabular structure equivalent to a standard spreadsheet, where the columnar elements correspond to the patient’s identifiers.

In contrast, row-wise elements correspond to the respective information of the patient identifiers. The identifiers link the tables, usually represented by the suffix ‘ID’. The table also uses the prefix operator ‘D’, which facilitates logical meaning toward the target identifier (e.g., dictionary).

The patient stays: This is a unique table meant to signify the information associated with the patient’s stay within the healthcare unit facility. Different information is associated with patient admission, approval for discharge of the patient, length of ICU stays, discrete information about the patient, clinical services offered to the patient, and transfer from one facility within the healthcare unit. It should be noted that there is a significant link to this table. For example, the identifier for ICU stays is linked to a single identifier of hospital admission and a patient’s identifier. Simultaneously, the patient’s identifier can be linked to different hospital stay identifiers and ICU stays.

Critical care: This is another unique table comprising information about the patient’s stay in an intensive care facility. It bears information about the identifiers of caregivers with all events marked in the respective medical charts. It also possesses information about events, time, the date for any medical procedure, device-based patient intake, and monitoring. Apart from this, it has various de-identified data related to radiological reports, ECG reports, physician notes, a summary of discharge, patient output during their stay in ICU, and all other details on procedures being performed on the patient during the patient’s stay.

Hospital record system: This table comprises information associated with the data recorded within the hospital management. It consists of information about various procedures being carried out with the inclusion of codes and information about the diagnosis stated by the hospital as per the standards of ICD. It also consists of various diagnosis-related information especially needed for the billing cycle. Different laboratory results measurements, microbiological test information, ordered medication, and procedures for patients are coded and stored in this table according to ICD.

Dictionary-inclusive table: Dictionary-based information offers extensive information about the field in a table. It consists of high-level codes associated with current procedural terminology. It also has a dictionary about ICD connected with procedures, different inclusions of MIMIC items, and information about laboratory tests. Regardless of a higher degree of adoption and utilization of the MIMIC-III dataset, it remains challenging, mainly when predicting trajectory. The prime reasons behind it are:(i)Adopting highly granular standards, such as ICD-9, and(ii)Reducing cardinality. The cumulative quantified admission is discovered from the cardinality perspective.

It is 58,976, with a visibly skewed number of admissions considering the patient distribution. That can be observed in Figure 1, where one admission consists of 38,983 patients [18]. Based on this fact, it can be said that MIMIC-III consists of patient data where only two entries can be considered fruitful toward the prediction of trajectory. Many associated ICD-9 codes are absent from the dataset values, while other matters are illogical with negative time duration.

Figure 3 shows multiple patients with multiple admission numbers while a distinct set of ICD-9 codes represents each patient. The MIMIC-III dataset maintains the data based on hospital admission number, which is uniquely allocated even for the same patient having multiple visits. Hence, analysis is based on hospital admission numbers of the computationally expensive process with the current state of using the MIMIC-III dataset. From this perspective figure within the dataset, the number of admissions comes somewhere around 19,911, while the patient number comes to about 7683 from 46,520. From the ICD-9 encoding system viewpoint, MIMIC-III is characterized by another pitfall associated with a higher number of standards of ICD-9 that offers a code of diagnosis equal to 15,072. It is the reason that ICD-10 provides four times more information. This granularity of the details is represented by the cardinality that further explains the illness connected with clinical trials’ feasible manifestation. There are cumulative 6984 codes in the MIMIC-III instance, where Figure 4 shows the analysis of patients with respect to hospital admission over time comes around 7683, while Figure 5 highlight the code distribution concerning quantified hospital ad-mission to have a total of 911 from 46,520. Mathematically, this distribution cannot be termed a Gaussian distribution due to the inclusion of outliers associated with the admission of nine codes. However, due to the proximity of the Gaussian distribution, the researchers can consider the normal mean of access with 13 codes as the feasible parameter of descriptive statistics. This leads to the conclusive remark of the predictive task for the codes associated with consecutive admission to lying beneath 9.3 × 1049 feasibilities.

As a result, a higher degree of granularity is a significant issue in ICD-9, and the effect of this problem is far-reaching in different investigations. This problem is reported to be addressed by using the encoding mechanism of Clinical Classification Software (CCS) [19]. This classification policy determines a tabular mapping of the specialized form using ICD-9 to obtain minimal description in granularity standards. The goal is to assist in the faster and easier analysis of medical data and an efficient reporting system. Therefore, the usage and adoption of CCS result in a consecutive admission score of 3.2 × 1031 feasibilities, right from 9.3 × 1049 feasibilities. Usage of CCS encoding system will eventually reduce 18 orders of measurement feasibilities. This fact also contributes to prediction performance owing to its simplification. Unfortunately, although this approach reduces granularity, it also has the disadvantage of introducing minimal details that affect the predictive and descriptive outcomes. It will also potentially impact classification performance and decision-making. Another difficulty is that such low quantities of 19,911 samples are highly insufficient to deploy neural network training algorithms if subjected to 13 codes. Hence, the MIMIC-III dataset opens the door to additional research opportunities to make it more practically implementable in clinical analysis.

## 4. Predictive Approaches for MIMIC-III

To obtain a precise prognosis of any lethal medical condition, the system must enable accurate sequential processing of the patient’s complete information [20]. Processing the patients’ set of sequential event-related information also helps formulate compact data that could further help develop a prognosis [20]. However, creating a precise predictive operation will require overcoming multiple challenges where the primary issues arise from aggregation and medical data management. Inconsistent, inappropriate, incomplete, and heterogeneous data will adversely affect the predictive operation [21]. An effective predictive process will require determining all the significant data elements depending on the potential utilization value. Increased usage and adoption will further facilitate better predictive operation. As a result, this section explores the existing approaches toward leveraging an efficient predictive strategy considering the MIMIC-III dataset.

### 4.1. Classification-Based Approach

The classification process is generally associated with effectively organizing the medical data in logical categories to assist in diagnosis. The International Classification of Diseases (ICD) is considered a standard body of classification systems for multiple acute disease variants [22]. The classification problem must be addressed to carry out the predictive operation. Studies towards improving classification also assist in predicting the phenotype classification [23], the mortality rate in hospitals [24], and assessing decompensation of psych [25]. It also helps evaluate the length of stay by using multiple and binary classifiers to measure the risk factors associated with a more extended stay.

Table 1 highlights various available classification-based approaches. The work carried out by Shi et al. [26] has developed a process where the diagnostic codes are autonomously assigned. Conventionally, ICD-9 codes are allocated by the system (man/machine) based on short names of the diagnosed disease or procedure in the record, which is tedious and sometimes time-consuming. However, this work is different from the conventional approach. It works on independently allocated diagnostic codes without depending upon any system. The study generates the ICD codes with latent representations using deep-learning data from thousands of hospital admissions. The study obtains the confidence value of the ICD score using dual attention layers, using both hard and soft selection mechanisms. The computation of the *i*th code of ICD probability is carried out as follows:(1)$$\;prob_{i}=f\left(j=1,2,\ldots arg_{max}\left(a_{i,j}\right)\right)$$

In the above expression (1), the function *f*(x) represents a sigmoid function for normalizing probability. In this expression, the attention score *a_ij_* for *i*th code of ICD and *j*th information about diagnosis considered over *U* dimension of data and *h* hidden representation of the *k*th size. Therefore, the cumulative expression of attention score *u* is given as [26],
(2)$$\tilde{u_{l}}=\sum_{j=1}^{m}\frac{\exp\left(\sum_{k=1}^{d}u_{i,k,h_{j,k}}\right)}{\sum_{j=1}^{m}\left(exp\sum_{k=1}^{d}u_{i,k,h_{j,k}}\right)}.$$
where expression (1) and expression (2) represent a hard and a soft score of attention, the study outcome exhibits a better performance score of the soft attention layer in deep learning than the hard attention layer over multiple diagnosis evaluations. However, the model accuracy could have been further enhanced by applying preprocessing over the data that has not been carried out, which is currently found in a few works of literature [27]. A similar line of research work has also been carried out by Li et al. [28], Nigam [29], and Kassis [30], considering ICD-9, where the codes are automatically assigned. This technique integrates the document-to-vector approach with a convolution neural network to obtain cumulative features (both local and global) (Figure 6).

Another supervised learning approach resulted in a 14% performance improvement. Although the study supports classification with multiple labels connected with patient notes, more high-quality data can improve the learning task for better classification performance. Bao et al. [31] recently worked on a predictive scheme on MIMIC-III data using a capsule network, where the authors have emphasized the benefits of adopting a deep learning scheme. Sagi et al. [32] carried out a similar pattern of work. Ye et al. [33] used Natural Language Processing (NLP) and machine learning for mortality prediction considering the case study of diabetes. Hou et al. [34] used XGboost for mortality prediction considering a case study of sepsis-3, demonstrating a better predictive scheme in its outcome when compared to a conventional regression scheme. Apart from this, only a few studies have improved the classification-based approach over the MIMIC-III dataset.

### 4.2. Numbering and Spacing

Apart from the classification-based approach, various predictive-based operations have evolved over the MIMIC-III dataset to assist in an effective prognosis (Table 1). Existing studies emphasize phenotyping analysis and predict the risk of mortality associated with intensive care unit data (Veith and Steele [35]). The present research has witnessed the usage of the predictive approach. The existing system has presented predictive modeling work considering multiple electronic healthcare information sources (Gong et al. [36]). This approach constructs features from the MIMIC-III database using text analysis and a bag of events (Figure 7). According to Figure 7, the model discusses the classification of two forms of stroke, i.e., ischemic and hemorrhagic stroke. The distinctive features of the above strokes are further classified into various minor levels of bag of words. The core idea is to simplify the classification process to facilitate better diagnosis over MIMIC data based on text. The study was used mainly to assess the in-hospital mortality rate and length of stay.

Most of the predictive operations performed on disease prognosis have focused on the length of stay in the hospital, which is essential for resource planning (Rojas et al. [37]). The neural network-based approach is proven to be highly dominating in this direction. Gentimis et al. [38] used a simplified process where the WEKA tool carried a selection of specific attributes of MIMIC-III data. Information associated with admission, events, stays, services, patient details, procedures, and diagnosis is used for this analysis. This information about multiple organ systems is considered (Chen et al. [39]). It is seen that the adoption of the recurrent neural network over deep learning plays a contributory role in this regard. This solution uses a time-series representation of each organ’s physiological attributes, and a learning approach has been developed considering the specific task. The study uses Long Short-Term Memory (LSTM) to derive the co-relationship among different tasks. The predictive outcome is improved using an attention mechanism over the implemented deep learning approach. The study outcome directly correlates prediction performance with increasing time windows. Simultaneously, the proposed system (MTRNN-ATT) exhibits better accuracy than other existing predictive approaches (Figure 8). The analysis considered 80% of a random selection of ICU data of 26,740 stays to find lower accuracy performance for a recurrent neural network with multitasking (MTRNN) without using an attention mechanism to that of using an attention mechanism (MTRNN-ATT).

Prognostication during ICU is often studied using linear models that cannot analyze temporal trends. The existing system has been reported using a machine learning approach to solve these issues where predictors using non-linear attributes are developed (Meiring et al. [40]). Using machine learning, the mortality trend in the ICU is studied with respect to temporal parameters, and it is observed that logistic regression enhances the accuracy of the outcome over physiological data. The study has also implemented a support vector machine, AdaBoost, random forest, and radial basis function. The study, however, needs to emphasize the missing data problem, and the considered dataset is reported to be unbalanced.

Aside from the length of stay, the current predictive approach is also used to forecast prescriptions for the upcoming period (Jin et al. [41]). This study has used the conventional LSTM model of the neural network, divided into three categories: decomposed, partially connected, and fully connected LSTM model inclusive of heterogeneity. Using different attributes, e.g., age, gender, time, lab indices, and prescription, this model helps generate a comprehensive treatment plan using a recurrent neural network. The model is also assessed for several prognoses with higher accuracy as compared to conventional LSTM models. Xia et al. [42], Yu et al. [43], Li et al. [44], and Yang et al. [45] too have published additional research using an LSTM-based approach. The prime distinction among these studies is mainly associated with usage of the LSTM approach. LSTM has been integrated with an ensembled algorithm for Xia et al. [42] work. In contrast, LSTM is combined with latent semantic analysis for encoding in Yu et al.’s [43] model. Li et al. [44] have used the directional nature of LSTM, and Yang et al. [45] have used LSTM with time-series embedding.

There is no doubt that predictive studies using recurrent neural networks have been proven successful in the clinical prediction of mortality and phenotyping. However, such approaches also heavily depend on massive labeled data during their operation. Apart from this, it is also stated at the beginning of this section about the need to maintain consistency, appropriateness, and effective structurization of clinical data using machine learning. This is where there is a demand created for applying preprocessing operations over any existing machine learning approaches on clinical datasets, as stated in the work of Aljuffri et al. [46], Zelaya [47], and Fangyu et al. [48]. In this perspective, a recurrent neural network is found to sort out this problem to some extent and is quite applicable in learning-based data structurization where current computation is dependent on prior outcomes. As the recurrent neural network can process sequential data, its suitability for healthcare analysis is relatively high. Such issues in predictive modeling can be addressed by pre-training and fine-tuning using transfer learning (Gupta et al. [49]). The performance of the recurrent neural network can be leveraged by pre-training over massive time-series data distribution. Adopting such an approach offers independence from the manual selection process of features with supportability towards using linear models. This model’s excellent idea is to introduce a unique architecture (Figure 9) for encoding and decoding using three layers of recurrent neural networks.

It also includes extracting features using weights and time series while extracting relevance scores that increase accuracy for adopting raw input features. Further, Figure 7 also highlights the mechanism of applying time series to the basic features, followed by embedding using an encoder of a recurrent neural network to obtain the final feature and weight. This accepted feature is used for performing prediction and using relevance scores to elevate the accuracy level to a clinical perspective. Rodrigues-Jr et al. [50] and Su et al. [51] completed additional research using recurrent neural networks. Some research has been done on critical disease prognoses, such as sepsis. It is the most challenging disease to detect; thus, it often poses issues in performing prognosis. However, the machine learning approach has already been proven to resolve this issue. Mao et al. [52] recently used a gradient tree boosting method for sepsis prognosis, considering six vital signs. The implementation offers higher accuracy in prognosis and is not affected by lacking data.

The study is implemented over mixed ward data, in which the model can identify and predict different variants of clinical sepsis states. According to the authors, adopting the ICD-9 standard is one prime limitation due to its restriction on the investigation. Apart from this, the research also considers the inclusion of unnecessary non-physiological attributes. However, the study contributes to the state’s better possibility of enhancing prognostication if subjected to delay.

Further studies on sepsis have been carried out by Scharpf et al. [53], where the recurrent neural network has been used and where the study performs ranking of the implemented classifier. This technique is reported to extract 101 features where the mean conditional probabilities are considered from the descriptive statistical information of each parameter’s numerical score. This parameter contributes to higher possibilities of sepsis considering various parameters of a patient’s clinical data. A similar direction of work toward sepsis prognosis is also carried out by Desautels et al. [54]. The adoption of the recurrent neural network is also witnessed in Xu et al. [55], which addresses the challenges of processing high-density signals in multi-channel and multiple data modalities. A model is constructed to monitor physiological data of various channels with rare clinical events. The recurrent attentive model is further developed for processing the multimodal inputs where LSTM is applied to obtain the outcome with better interpretability.

Apart from sepsis, the current study has also been carried out on cardiac arrest prognosis, which has a higher mortality rate (Nanayakkara et al. [56]). The investigation has been carried out using logistic regression, random forest, support vector machine, ensemble classification, and neural network over multiple vital signs of cardiac arrest. According to the author, the predictive performance is better for deep learning than logistic regression and other statistical models. It also suggests that the explainability of the prognosis is further improved when deep understanding is integrated with explainer attributes (Figure 10). The scheme exhibited in Figure 8 introduces various use cases of introducing logistic regression over the data, whose accuracy score is not much improved owing to its assumption on linearity existing between dependent and independent variables. This problem was sorted out when varied machine learning models were hybridized with the statistical model, as seen in the second block of operation. However, the authors discussed the non-dependency of feature engineering being one of the prime benefits of deep learning methods compared to machine learning. Hence, predictive accuracy can be further increased when the deep learning approach is used with an additional module called Explainer, which adds more information and contextual content for the data during prediction. However, there are still issues associated with clinical translation using conventional datasets.

Predictive prognoses associated with bloodstream infection are also being studied (Parreco et al. [57]). It has been discovered that supervised machine learning could provide better predictive performance in such cases. Another frequently used predictive scheme is document embedding using a neural network (Grnarova et al. [58]). A two-layered architecture is said to be used, with the first layer mapping sentence vectors to individual sentences and the second layer integrating all the sentence vectors to offer an integrated patient representation.

Figure 11 highlights the architecture where word-level CNN is designed to consider s1 sentences of patients, thereby generating x1 as a sentence vector. This is repeated until an xn sentence vector is developed, taking into account n total sentences to generate x as a patient vector. Finally, the probability of mortality *p* (*y*| *X*) is computed to lead to the cross-entropy, i.e., (*y, y*| X*), where the ground truth data is represented as *y** along with the computation of the mean error of prediction, i.e., R with an objective function to minimize label L with regularizer strength λ. The study outcome performs better than other document embedding approaches, e.g., document to vector and bag of words. The machine learning approach, such as extreme gradient boosting, has offered a predictive analysis of patients suffering from acute kidney illness (Zhang et al. [59]). In addition to neural networks, fuzzy logic has also contributed to effective prognosis using the MIMIC-III dataset. Davoodi and Moradi [60] conducted a unique study using deep learning and enhanced fuzzy logic to analyze vast medical data. According to the authors, traditional fuzzy logic is difficult to implement when processing and analyzing large amounts of medical data. Thus, only the improved version of fuzzy logic possesses this capability. Consequentially, it can be noticed that authors have various predictive approaches for improving the accuracy of prognosis.

### 4.3. Early Predictive Approach

A certain degree of work within the predictive analysis domain emphasizes the early detection and prognosis of clinical data (Table 1). Ding et al. [61] used an artificial neural network to predict acute pancreatitis using the MIMIC-III dataset to develop an early predictive approach. More recently, multivariate logistic regression has been proven for an early prediction approach considering multiple physiological scores (Zimmerman et al. [62]) over the MIMIC-III database. More care is needed in ascertaining the system model’s reliable performance in performing an early prediction where sustainable performance is demanded. The recent modeling presented by Lin et al. [63] has discussed random forest usage. Multiple predictive models (e.g., support vector machine, neural network, tailored SAP II) were studied, and numerical outcomes were assessed with a Brier score. The study outcome shows that random forest has better accuracy in early prediction than existing models. However, the calibration performance of the random forest is similar to the support vector machine. Therefore, the study outcome suggests that each learning model has its scope and limitations. From the interpretability viewpoint, the author advocates using random forests compared to support vector machines. Random forests’ early predictive models offer better decision-making because they are ensemble classifiers. Simultaneously, the support vector machine’s outcome could be more readily understandable from a user-friendly viewpoint. Hence, the study encourages random forest usage to predict in-mortality rates in hospitals.

Nestor et al. [64] showed that existing approaches are not benchmarked and cannot be generalized. The investigation reveals the degradation of the prediction quality when historical data is trained while testing is carried out over future data. Such a model improves accuracy while working on an early predictive approach. The early predictive process is also essential for considering a heterogeneous group of data (Suresh et al. [65]). A study on early prediction has been carried out by Li et al. [66] that has used natural language processing and machine learning. The study has used text-based information processing, extracting features via a bag of words, followed by multiple machine learning algorithms (gradient boosting decision tree, random forest, logistic regression, and Naïve Bayes). The current study has also used vital signals to detect mortality rates (Sadeghi et al. [67]). The study derives the heart signals from multiple statistical features subjected to various classifiers.

Further, a unique study has been presented recently where early prediction has been carried out using machine learning (Javan et al. [68]). This work is about the prognosis of detecting secondary fatal illness owing to the primary disease. For this purpose, multiple features, e.g., time series, multivariate features, and latent clinical features, are used from the MIMIC-III database to determine the exact time interval for identification. The ensemble learning approach is also reported to assist in early detection (Awad et al. [69]). Figure 12 highlights the mechanism that includes data preprocessing followed by attribute selection considering two distinct forms of attributes, i.e., missing data and class imbalance, to construct a model for early prediction.

The work by Garcia-Gallo et al. [70] has carried out a 1-year early prediction of sepsis considering 5650 admissions. The study has used a stochastic gradient boosting approach where satisfactory accuracy is found. However, the narrowness of the data is seen as it considers only unit institutions. The following section discusses the diagnosis aspects of the MIMIC-III database.

## 5. Clinical Diagnosis with MIMIC-III

To carry out an effective prognosis of MIMIC-III’s clinical database, existing approaches have also emphasized various associated perspectives to facilitate precise representation. This can be reduced by enduring the following conditions: (i) adequate interpretability, (ii) formulating a practical decision, and (iii) performing a proper evaluation of target data concerning the specified problem. The discussion of the existing approaches (Table 1) is as follows.

### 5.1. Disease Characterization Interpretation

As the MIMIC-III dataset contains enormous and various kinds of data inclusion, it is essential to select a suitable approach to interpret the data precisely. It is necessary, especially when the data is subjected to complex processing later. Bashar et al. [71] and Fan et al. [72] recently studied detecting atrial fibrillation and renal damage using the MIMIC-III dataset. Various data analysis tools extract disease-related information (Dai et al. [73]). The study contributes to obtaining informative details from a large dataset and characterizing MIMIC-III. Another significant issue is understanding the clinical conclusion of the diagnosis. This problem is addressed in Prakash et al. [74], where a memory network has been used to process complex tasks.

All the significant features extracted are stored in a condensed memory system using a neural network and, thus, help in diagnosis clarification. Due to the increased possibility of disambiguation in medical words, such studies demand its operation to support distributed representation of medical data. Such issues are solved by combining medical word representation related to a sizeable textual body with less standard definitions provided by the Unified Medical Language System (UMLS). (Tulkens et al. [75]). However, the outcome shows variations in inaccuracy when the word corpus is changed. One of the primary issues with such clinical notes is the usage of medical codes that are not explicitly considered in this study. Even if such clinical notes are considered, the annotation will become tedious and error-prone.

Incorporating a convolution neural network and attention mechanism is proven to help choose the most appropriate segments, improving interpretability (Mullenbach et al. [76]). The outcome was better than conventional bag-of-words, recurrent networks, and single-dimension convolution neural networks. From the discussion stated in the prior section, it is obvious that the machine learning approach is dominant, especially among all variants of the neural network approach. All the approaches towards classification and predictive-based operation need to emphasize the interpretability of its outcome from a clinical perspective. Another impediment to interpretability is the temporal dependencies and inclusion of high-dimensional data. This problem is solved using a gated unit’s recurrent neural network (Sha and Wang [77]). A hierarchical attention model is constructed (Figure 13), which takes the medical record in chronological order and encodes the ICD codes using bidirectional gated recurrent units, which significantly maximizes input information.

Furthermore, the study incorporates an attention mechanism at the code level, where additional attention is provided over ICD codes. The system also includes visit-level attention to improving interpretability in its prediction process. Diagnostic codes are used to validate such a model. Interpretability and characterization of the data also depend upon the medical data’s completeness, which cannot be ensured. Accordingly, with the adoption of advanced cloud-based medical data management applications, there is always a possibility of missing data, significantly affecting the accuracy and interpretability of results. This issue can be addressed using bidirectional gated recurrent units (Che and Liu [78]).

The study used soft labeling learning using deep neural networks to filter the knowledge from MIMIC (Figure 14). However, the study did not address the data insufficiency associated with deep learning usage. This issue can be addressed by constructing an attention model using a graph (Choi et al. [79]). The medical concepts associated with the higher level are generalized to lower levels to address this issue using recurrent neural networks. The study outcome finds that the presented approach offers better interpretability than existing, recurrent neural network models using end-to-end training.

### 5.2. Detection-Decisions Approach

An accurate and efficient predictive approach and classification are only possible if the model incorporates a better form of decisive logic. However, there are dependencies on including many precise clinical parameters. Existing approaches in some studies may highlight this aspect of the MIMIC-III database associated with decision-making heuristics. McWilliams et al. [80] have recently constructed an autonomous technique using the decisive logic construction method. This study advocates the usage of the decision support tool. Alon et al. [81] recently described a similar direction of work. The features are extracted from the MIMIC-III dataset, followed by multi-layer perceptions and evaluation of many classifiers over ICD-9 codes. However, one of this technique’s drawbacks is that it provides insufficient evidence for decision-making when the environmental information is dynamic. Chen et al. [82] used recurrent neural network multitasking where LSTM is used for the learning approach. Furthermore, LSTM is also used to draw connectivity between multiple tasks using an attention-based method. The approach provides a higher range of features for use in timely decision-making in the classification process. A similar system was also reportedly used by Kaji et al. [83].

### 5.3. Evaluation-Based Approach

The evaluation-based approach is primarily an exploratory mechanism for testing the introduced module’s effectiveness using various over-analyzed test parameters in the MIMIC-III dataset. In the current system, reinforcement learning facilitates quantitative treatment strategy evaluation (Raghu et al. [84]). Apart from the quantitative approach, the empirical evaluation approach has also shown its value in prognosis when ICD-9 codes are subjected to deep learning (Huang et al. [85]). The evaluation has also been conducted considering a comparative-based study by Gehrmann et al. [86], where deep learning and rule-based systems have been assessed. The outcome shows that deep learning can extract better information for prediction than other phenotypes. A similar comparative approach was also discussed by Beaulieu-Jones et al. [87], where trajectories of patient data are mapped with deep learning. The prognosis model is also required to offer reliable outcomes, demanding an effective benchmarking strategy (Purushotham et al. [88]). Harutyunyan et al. [89] conducted a study in this direction, where benchmarking was performed on the multitasking learning mechanism over the MIMIC-III dataset. This approach presented potential neural and linear baselines considering multiple clinical detections, classification, and prediction issues. Apart from this, there have been various other approaches that have completely used evaluation-based methods, viz. Meyer et al. [90], Islam et al. [91], Roehrs et al. [92], Su et al. [93], Jonson et al. [94], Shafaf and Malek [95], Zeng et al. [96], etc. All the techniques used in the above-mentioned work help offer a constructive guideline for identifying dependable factors for an effective prognosis system.

## 6. Learning Outcome of the Study

It is to be noted that the mode of discussion being carried out in this current paper is mainly on different forms of scientific techniques towards the MIMIC-III dataset, which contributes to disease prediction and diagnosis. However, in this direction of methods, a particular essential contribution has been emphasized evolving with novel and efficient clinical research techniques but does not include the MIMIC-III dataset. The research article that has been come across in this perspective is briefed as follows:

The significant research article presented by Peek and Rodriques [97] and Lausser et al. [98] has discussed potential problems with analyzing medical data. The recent work by Redondo et al. [99] has used a machine learning approach to assess the effectiveness of a priori rules over different patterns of stages of cancer. The adoption of machine learning for identifying critical medical conditions is discussed in Piadlo et al. [100], which states that there is still a large gap between the demand for clinical diagnosis and existing model effectiveness. Another recent study model by Sheth et al. [101] used a cognitive approach to construct a chatbot technology for elevating the communication of patients using Internet-of-Things. A unique non-clinical-based study was carried out by Lu et al. [102], where the idea is to understand the set of anomalies in claims of medical insurance. Chen et al. [103] have presented a wearable technology to promote the autonomous transmission of clinical data using federated transfer learning. The adoption of deep learning has been uniquely implemented by Pabon et al. [104], where clinical information is extracted from the text of specific linguistics. The adoption of the Attention Neural Network is seen in the work of Oba et al. [105], which contributes to the prediction of diseases based on regular health records. From the diagnosis perspective, the existing system has also witnessed studies on decision-making (Kokciyan et al. [106]). A closer look into the above current studies shows that techniques are more or less similar in machine learning and deep learning practices, with the only difference of non-adoption of the MIMIC-III dataset. Hence, various conclusive remarks are found after analyzing the existing approaches to the predictive scheme. This study has explored various essential techniques for analyzing the MIMIC-III dataset. Table 2 compares different methods systematically based on MIMIC-III concerning advantages and limitations associated with core implementation-based articles [28,29,30,31,32,33,34,35,36,37,38,39,40,41,42,43,44,45,46,47,48,49,50,51,52,53,54,55,56,57,58,59,60,61,62,63,64,65,66,67,68,69,70,71,72,73,74,75,76,77,78,79,80,81,82,83,84,85,86,87,88] only, while remaining articles either discuss concept or do not offer significant insight. Based on these observations, a judgment can be made concerning current approaches.

## 7. Open-End Issues

Prior to understanding issues in existing research work in MIMIC-III dataset, it is to be noted that its updated version, i.e., MIMIC-IV, is also available for public usage. In fact, the MIMIC-IV dataset offers a much more modular and structured form of clinical data with proper linking of data modalities and database to external department [107]. However, a closer look into the trend of usage of MIMIC-III and IV version shows that various significant modelling attempts have already been established in MIMIC-III extensively compared to MIMIC-IV. So, the concern is towards improving the performance of all the models that have been testified with issues when experimented over MIMIC-III dataset. There is no major difference in both the datasets as problems still exist to some extent, even in the MIMIC-IV dataset in perspective of its structuredness that is demanded for performing advanced analytical operation. Hence, the present focus is mainly towards its legacy version which is extensively deployed. While some of the remarks are related to the practical perspective and others to the limiting factors, the valuable view of the current research work is already stated in the literature, and therefore this section briefs about limiting factors of the existing research approach as follows:

Inadequate focus on preprocessing: The majority of work carried out over the MIMIC-III dataset has not considered the fact that there could be a possible presence of artifacts in the data [71,72,73,80,85,87]. Such artifacts could be missing data, surplus data leading to redundancy, noisy data, and impartial data. The reasons for such artifacts could be many, from distributed data storage to massive data transmission. Hence, once these artifacts are subjected to preprocessing, predictive outcomes can be ensured with higher accuracy by considering the practical constraints.

Computational complexity is yet to be studied: MIMIC-III data has higher granularity and high-dimensional data. Processing such massive data to solve a specific query system demands faster processing, whereas all existing approaches emphasize accuracy [26,27,28,29,30,31,32,33,34]. When it comes to critical prognosis, a time-bound operation is required. Hence, there is no consideration of faster algorithm processing time or lower memory consumption while processing the iterative machine learning algorithm found in the existing predictive scheme.

Minimal constraint inclusion: Most of the work carried out in the predictive approach focuses on one set of problems. Many of them are more inclined to detect [71,102], classify [26,27,28,29,30,31,32,33,34], and predict one set of predefined diseases [35,36,37,38,39,40,41,42,43,44,45,46,47,48,49,50,51,52,53,54,55,56,57,58,59,60]. Various extrinsic and intrinsic parameters also control the predictive modeling where uncertainty is not included. This non-inclusion of essential parameters and reduced constraints will deem the model more theoretical and less practical.

Reduced scope of implementation: From the prior sections, it can be seen that there are various taxonomies of predictive approaches in MIMIC [61,62,63,64,65,66,67,68,69,70]. However, no such system ensures customized decision-making support [80,81]. The modeling is anticipated to include the demands of data acquisition, dimensional reduction, modeling perspective, prediction, and interpretation. All the existing approaches are more or less targeted at solving only predictive-based factors [84,85,86,87,88,89,90,91,92,93,94,95,96].

The need for a precise multivariate approach: None of the existing multivariate analyses [62] deal with stability issues, which is one critical demand for predictive modeling; they also possess a collinearity issue, potentially affecting the classification performance. Moreover, deep learning usage is still developing, and more specific implementation is further required with benchmarking.

Based on the study and findings of the proposed review, the following interpretations are as follows: (i) the review infers that the adoption of machine learning, as well as deep learning, has a different impact on the analysis process, especially in the perspective of the features; (ii) the outcome of the review suggests that there is a need to assess the inclusion of the various processing of MIMIC-III dataset before subjecting it to any analytical process; (iii) the complete review has encapsulated all major predictive schemes towards the MIMIC-III dataset as well as all clinical diagnosis schemes; and (iv) the study outcome potentially assists the upcoming researcher in ascertaining the selection of multiple predictive techniques to achieve these objectives of analyzing MIMIC-III dataset.

## 8. Conclusions

This paper has presented a compact discussion of the various predictive approaches implemented over the MIMIC-III dataset and its clinical diagnosis. The contribution of the study is in terms of the significant findings, which are as follows: (i) almost all prediction research is conducted using the ICD-9 code standard, which already has a granularity problem. There is no analytical modeling in any form to overcome this issue. (ii) Machine learning, which has dominated predictive modeling on the MIMIC-III dataset, frequently employs a convolution neural network, which suffers from overfitting issues and class imbalance. This potentially affects the deep learning approach’s prognosis mechanism, and the problem remains unaddressed. (iii) Few studies use MIMIC to predict secondary disease progression from primary disease, and almost all existing ideas prioritize MIMIC prediction over primary disease prediction. This will significantly lead to possible outliers in cancer-related patients where there is often a metastasis from primary to secondary illness. (iv) Various open-ended issues are associated with an existing prediction-based approach that does not support customization of decision-making and various other cases, as depicted in the prior section. The limitation of the study is that it does not present any form of discussion towards the data quality and impact of various clinical attributes, and this poses a major challenge towards analyzing critical care patients on medical data.

Hence, all these findings from our study must be considered for any further attempt to evolve the computational modeling of the predictive framework using the MIMIC-III dataset. We identify one preliminary and essential issue where preprocessing has very little scope in the existing implementation. Hence, we consider it as the first step to be carried out in the direction of our future research. While developing a preprocessing framework using the MIMIC-III dataset, the emphasis will be on covering a maximum of open-end research issues.

## Figures and Tables

**Figure 1 healthcare-11-00710-f001:**
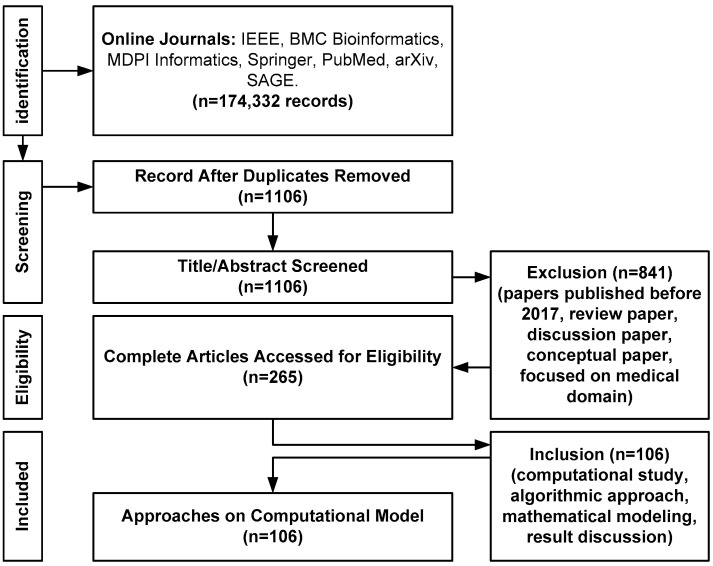
PRISMA Diagram for Proposed Systematic Review.

**Figure 2 healthcare-11-00710-f002:**
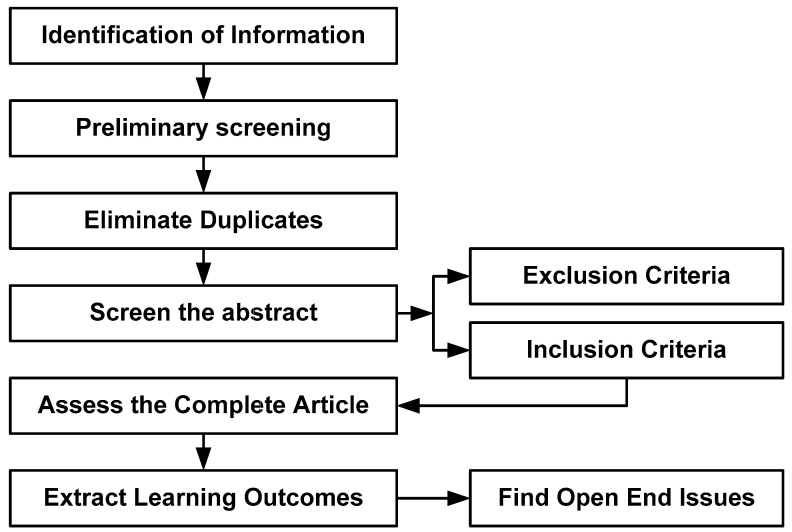
Flow of adopted PRISMA methodology of the current study.

**Figure 3 healthcare-11-00710-f003:**
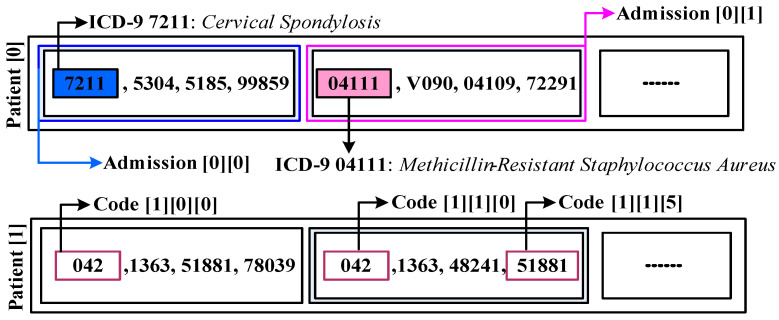
MIMIC-III with three levels of organization of records.

**Figure 4 healthcare-11-00710-f004:**
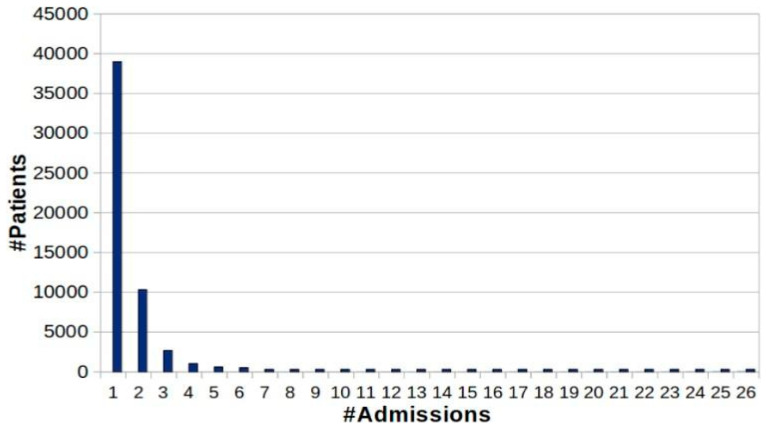
Patient distribution vs. admission number [19].

**Figure 5 healthcare-11-00710-f005:**
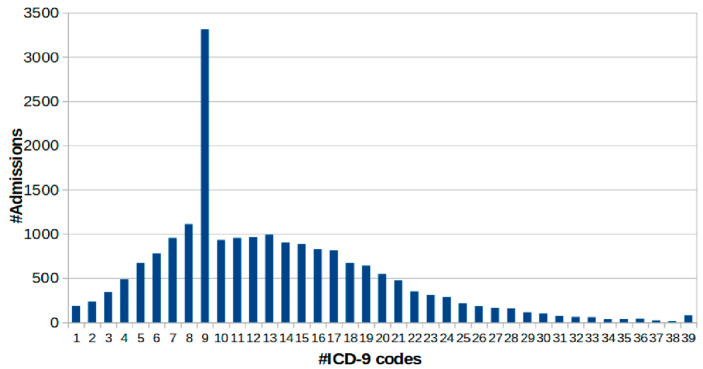
Admission distribution vs. ICD-9 codes [19].

**Figure 6 healthcare-11-00710-f006:**
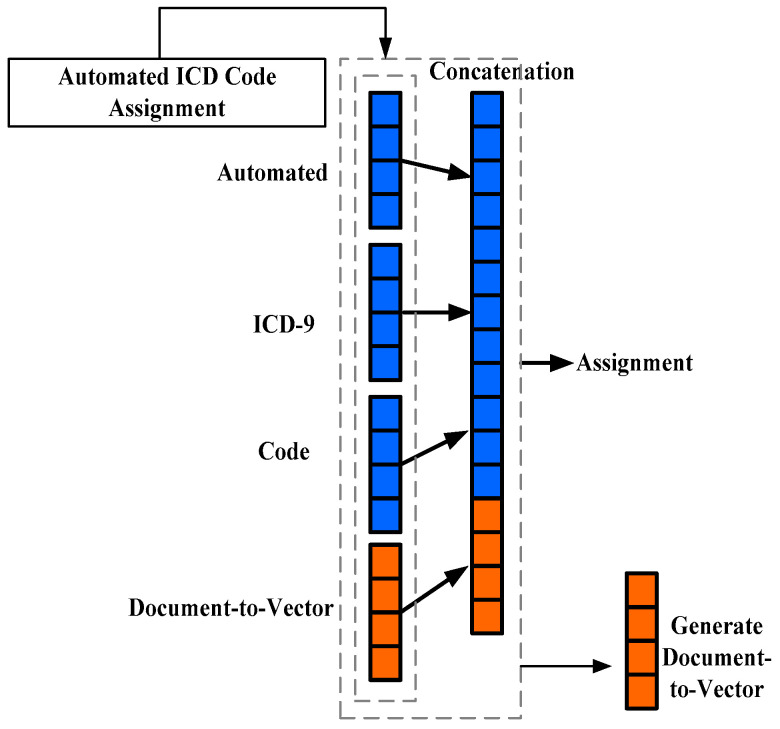
Generation process of document vector Li et al. [28].

**Figure 7 healthcare-11-00710-f007:**
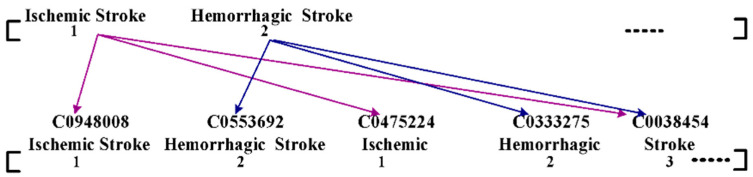
Adoption of Bag of Events (Gong et al. [22]).

**Figure 8 healthcare-11-00710-f008:**
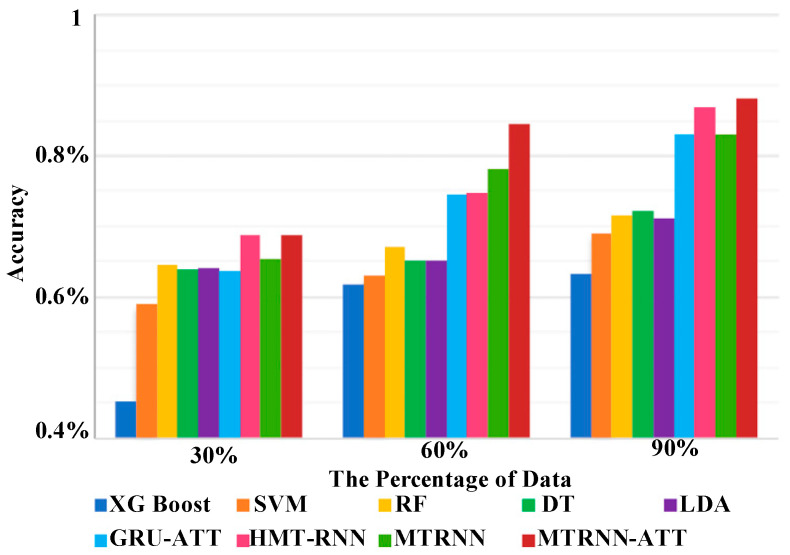
Outcome of predictive performance (Chen et al. [25]).

**Figure 9 healthcare-11-00710-f009:**
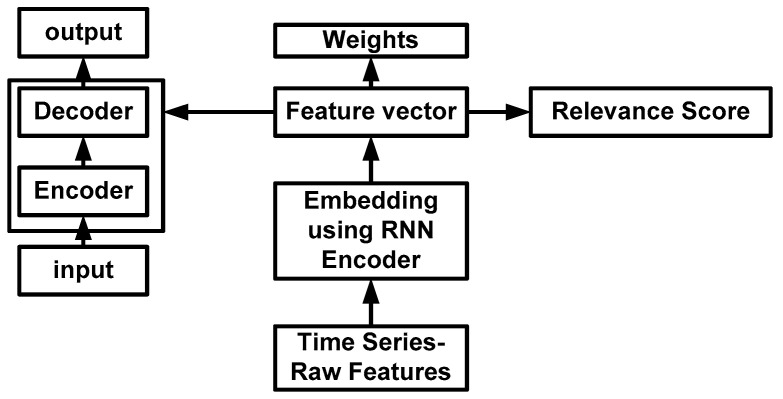
Architecture of recurrent neural network (Gupta et al. [32]).

**Figure 10 healthcare-11-00710-f010:**
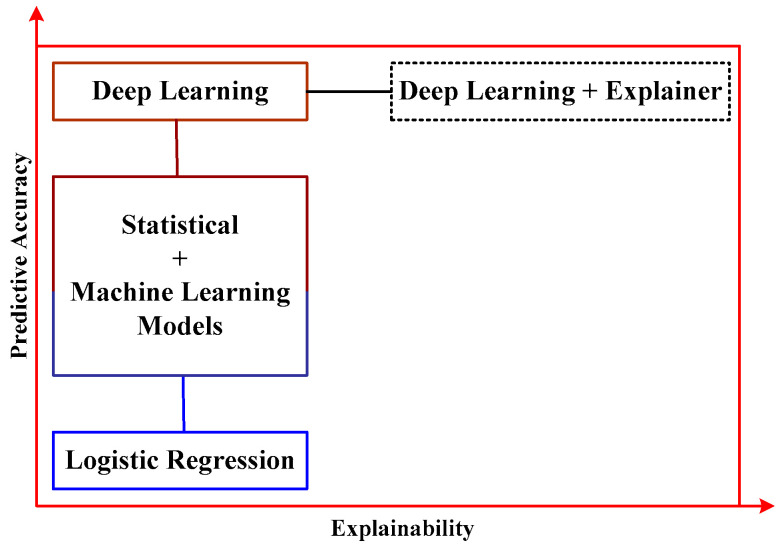
Performance of machine learning (Nanayakkara et al. [39]).

**Figure 11 healthcare-11-00710-f011:**
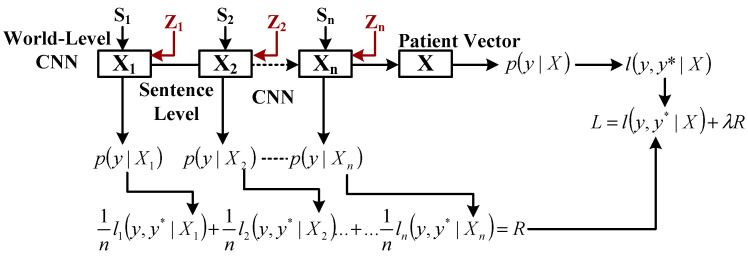
CNN-based predictive modelling (Grnarova et al. [41]).

**Figure 12 healthcare-11-00710-f012:**
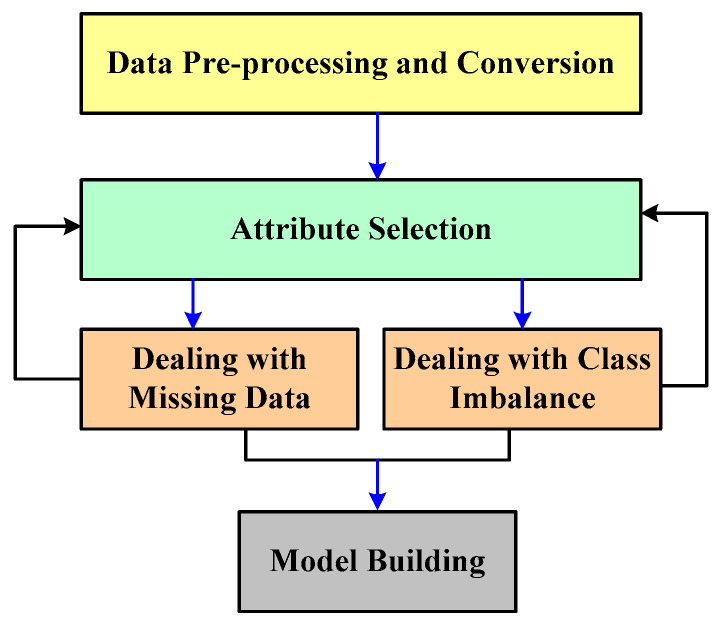
Early prediction process (Awad et al. [69]).

**Figure 13 healthcare-11-00710-f013:**
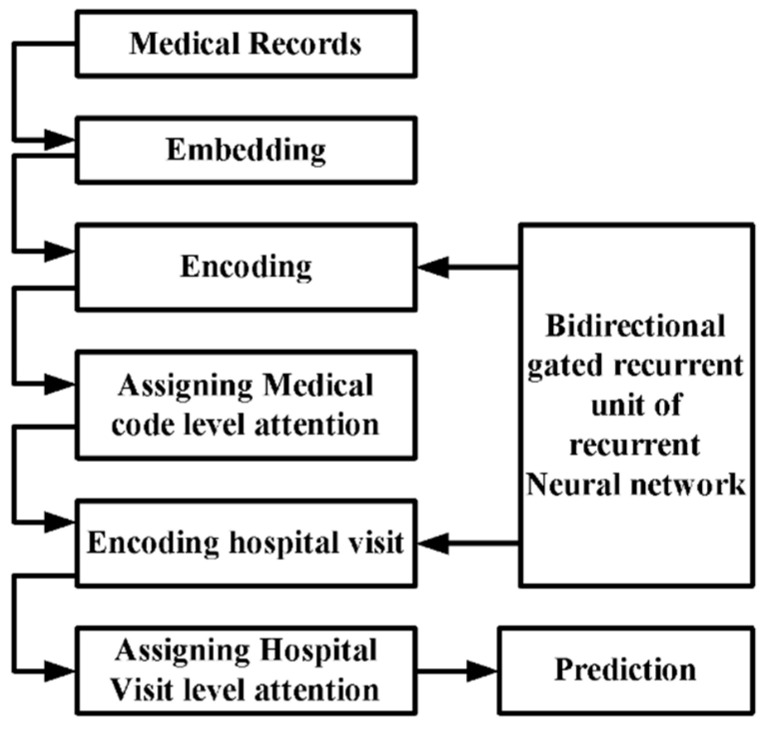
Architecture for predictive interpretability (Sha and Wang [77]).

**Figure 14 healthcare-11-00710-f014:**
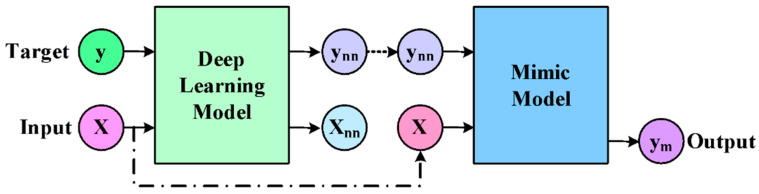
Learning of MIMIC (Che and Liu [78]).

**Table 1 healthcare-11-00710-t001:** Outcome of search result (2017–2023).

Bibliographic Database/Google Search Engine	Total Result	Final Result
IEEE	453	62
BMC	238	93
Elsevier	117	52
MDPI	4798	310
PubMed	13,552	402
arXiv	328	72
SAGE	4	4
Springer	4043	111
Total	23,533	1106

**Table 2 healthcare-11-00710-t002:** Learning outcomes of existing methods.

Scheme	Studies	Advantages	Limitation
Classification-based	ICD-based Li et al. [28], Nigam [29], and Baumel [30],	Practical analysis, when integrated with deep learning	It needs a massive number of data to come to the conclusion
	Capsule Network— Bao et al. [31]	No validation of medical studies yet	It is a slow-running algorithm
	NLP—Ye et al. [33]	A more straightforward analysis of medical data	Applicable for the specific task
	XGboost—Hou et al. [34]	Supports missing managing data	It does not offer scalability
Normal Predictive	text analysis and a bag of events-(Gong et al. [36])	Simplified and customized model for analysis	Specific to text data
	WEKA—Gentimis et al. [38],	Offers portability in analysis	Not meant for handling massive medical data
	Recurrent neural network (Chen et al. [39]), Rodrigues-Jr et al. [50], Su et al. [51]. Xu et al. [55],	Supports temporal-based prediction	Suffers from the complex training process
	Logistic Regression—Meiring et al. [40]	Simpler implementation without assumption	Linearity assumption
	LSTM model—Jin et al. [41], Xia et al. [42], Yu et al. [43], Li et al. [44], and Yang et al. [45]	Independent from finer adjustment	Does not address overfitting issues
	Integrated ML approach—Nanayakkara et al. [56]	Customized for the detection of various disease	Computationally complex process
	Document-Embedding & neural network, Grnarova et al. [58]).	Effective for all text-based medical report	Absence of benchmarking
	-Deep Learning & Fuzzy Logic, Davoodi, and Moradi [60]	Higher accuracy	Needs formulation of higher training epoch and a large number of rules
Early Predictive	ANN—Ding et al. [61]	The efficient and more straightforward training process	Fluctuation in training time
	-Multivariate Logistic Regression—Zimmerman et al. [62]	Effective for analyzing the relationship between parameters	Computationally expensive process
	Integrated ML approach Li et al. [66]	Customized for the detection of various disease	Computationally complex process
	time-series, multivariate features—Javan et al. [68]	Effective for temporal-based prediction	Requires higher iteration for training
Clinical Diagnosis	Detection of atrial fibrillation and kidney injury—Bashar et al. [71] and Fan et al. [72]	Progressive modeling toward disease analysis	Use-case specific
	Clinical conclusion of diagnosis—Dai et al. [73]).	Simplified inference system	Dependent on the type of dataset
	Neural network—Prakash et al. [74],	The efficient and more straightforward training process	Fluctuation in training time
	convolution neural network—Mullenbach et al. [76]	Autonomous feature detection	Slower processing
	Gated-recurrent neural network—Sha and Wang [77]).	Enhances the capacity of memory	Dependent on the specific form of medical data
	Characterization of data—Che and Liu [78]).	Adequate for concluding clinical inference	Dependent on the type of medical data structure
	recurrent neural networks—Choi et al. [79]	Memory efficient and offers scalable performance	It does not address the gradient vanishing issue
	decisive logic construction—McWilliams et al. [80]	Adequate for concluding clinical inference	Needs specific customization of data
	multi-layer perceptron Alon et al. [81],	Offers adaptive learning	Inclusion of a large number of parameters
	LSTM-Chen et al. [82], Kaji et al. [83].	Better control of network flow offers flexibility over output control	Does not address overfitting issues
	Reinforcement learning—Raghu et al. [84]).	Capable of solving a complex problem	Dependent on extensive computation resource
	Deep learning (Huang et al. [85]).	Practical learning of high-level features	Offers extra computational cost for training complex forms of data
	Comparative study—between deep learning and rule-based-Gehrmann et al. [86],	It offers a better evaluation platform	Not benchmarked with other variants of ML
	prognosis model—Purushotham et al. [88]	Simplified model of diagnosis	It does not draw a relationship among the essential clinical variables.

Table 1 showcases that each method has its beneficial features and limitations, which upcoming research initiatives must address. From the above discussion, it is essential to understand the significance of adopting the MIMIC-III dataset in different individual works carried out by authors. As stated before, the MIMIC-III dataset is a standard text-related dataset with clinical and non-clinical information about the disease diagnosed during the hospital stay. It will eventually mean that the MIMIC-III dataset introduces a fair set of computational complexity, rendering the dataset suitable enough to subject it to a learning scheme without much computational burden. Hence, the actual thing that matters is the techniques introduced by the existing research towards achieving a common target of disease diagnosis in the form of prediction and classification.

## Data Availability

The study uses the MIMIC-III clinical dataset, which is available on https://physionet.org/. Accessed on 25 January 2018.

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
