# Peer review of "Strategies of Predictive Schemes and Clinical Diagnosis for Prognosis Using MIMIC-III: A Systematic Review"

_healthcare, 2023, doi:10.3390/healthcare11050710_

Round 1

Reviewer 1 Report

Thank you for allowing me to review the manuscript entitled Strategies of Predictive Schemes and Clinical Diagnosis for Prognosis using MIMIC-III: a systematic review

The review is really interesting and the MIMIC-III program will give extremely valuable data  

My only comment is 

In the website of the MIMIC program they are know with the new version od the MIMIC (MIMIC IV) which also give insight of the pre-ICU management of the patients and this has not been addressed in the manuscript 

Reviewer 2 Report

1. PRISMA guidelines may be required to follow for systematic literature review.

2. Need to add methodology workflow diagram.

3. Search queries are required to retrieve articles from bibliographic databases, needs to mention.

4. The selection criteria of the journal are required to mention.

5. The inclusion and exclusion criteria must be described with keywords.

6. Tab. 1 caption not followed the journal template.

7. Article citation in the article does not match the reference list.

8. Fig. 5 is not visible properly.

9. Eq. 2 required rewriting and mentioning the source reference.

Reviewer 3 Report

The review is informative and useful. However, some of the figs are not clear; e.g., fig. 2,3,5.

Table 1 is too long. Present as supplemental  data.  

Reviewer 4 Report

Good job, but I think it needs some tweaking.

First of all, in the introduction, remove lines 77-82 since they don't make sense, and add the objectives of the study, at the end of the introduction.

In the methodology I miss a flow diagram of the choice of reviewed articles, as well as explanatory tables for each objective, the results are better interpreted.

The conclusion is too long.

The bibliographical references are not well placed in the text, for example in line 42-43, you should put [6-7] and not as it is, review the entire article.

Round 2

Reviewer 2 Report

1. Please add the following points for the search strategy

1.1 Do you have followed the PRISMA guidelines for search? Please mention.

1.2 Please mention the author name who was responsible for which bibliographic databases.

1.3 Please mention the year duration for which the primary search was performed. PubMed (1990- present)

1.4 avoid using random keywords for search; use some other option to add these articles to review.

2. Add a table showing the search results with a query for specific bibliographic databases.

3. Which software was used to remove duplicate articles/records?

4. How did you assess the risk of bias? Mention steps used to reduce it.

5. Write the complete form of CL given in the PRISMA diagram.

6. Please make a detailed PRISMA model.

7. Please mention the following points on the PRISMA diagram (separately each one of them)-

7.1 How many articles were excluded after reading the title

7.2 How many articles were excluded after reading the full abstract

7.3 How many articles were excluded after reading the whole body (paper)

7.4 Mention the reasons for article exclusion after full body reading.

8. In PRISMA Diagram 106, research articles were included (finally selected) for review purposes; only 38-40 articles were mentioned in table 1 (Learning outcomes). What about the others?

9. Please mention the limitations of the work.

Reviewer 4 Report

good job 

Author Response

The complete Manuscript is checked and edited by 

Dr. P.Saleema, Associate Professor of English, Aditya College of Engineering. Mobile-9912099892 [email protected]